# Cross-Contamination versus Outbreak: Pre-XDR Mycobacterial Strains Confirmed by Whole-Genome Sequencing

**DOI:** 10.3390/antibiotics10030297

**Published:** 2021-03-12

**Authors:** Jee Youn Oh, Kyung Ho Park, Jisoon Lee, Donghyeok Kim, Kwang Hyuk Seok, In-Hwan Oh, Seung Heon Lee

**Affiliations:** 1Division of Pulmonary, Allergy and Critical Care Medicine, Department of Internal Medicine, Korea University Guro Hospital, Seoul 08308, Korea; happymaria0101@hanmail.net; 2Division of Pulmonary, Sleep and Critical Care Medicine, Department of Internal Medicine, Korea University Ansan Hospital, Ansan 15230, Korea; book0811@naver.com (K.H.P.); blessing_ljs@naver.com (J.L.); 3Division of Bacterial Disease, Korea Disease Control and Prevention Agency, Osong 28159, Korea; kimdonghyeok@korea.kr; 4Department of Laboratory Medicine Center, Korean Institute of Tuberculosis, Osong 28158, Korea; stonelight@outlook.com; 5Department of Preventive Medicine, School of Medicine, Kyung Hee University, Seoul 02447, Korea; parenchyme@gmail.com

**Keywords:** laboratory cross-contamination, *Mycobacterium tuberculosis*, strains, whole-genome sequencing

## Abstract

Whole-genome sequencing (WGS) is promising for the quality control of laboratory facilities for *Mycobacterium tuberculosis* (MTB) strains. We describe the clinical and laboratory characteristics of false positive versus true positive MTB cultures based on WGS, which were experienced in a real clinical setting. Strain harvest and DNA extraction from seven isolates from pre-extensive drug-resistant (pre-XDR) TB patients transferred to the Korea University Ansan Hospital were performed, and epidemiologic links and clinical information, including the phenotypic drug susceptibility test (pDST), were investigated. WGS was performed using Ion GeneStudio with an ION530tm chip (average sequencing depth, ~100-fold). In the phylogenetic tree, identical and different strains were distributed separately. Five of the seven isolates were identical; the remaining two isolates differed from the others. The images of the referred pre-XDR-TB patients with false positive MTB that were analyzed were of regions close to old TB scars. Further, the results of WGS gene mutation analysis for ethambutol, streptomycin, and fluoroquinolone resistance in all six patients were not concordant with the pDST results. WGS and clinical information were useful in differentiating laboratory cross-contamination from true positive TB, thereby avoiding the unnecessary treatment of false positive patients and delay in treating true positive TB patients, with reliable genotypic drug resistance results.

## 1. Introduction

Tuberculosis (TB) remains a global threat, and its outbreaks have an enormous social impact [1,2]. Laboratory cross-contamination mimicking outbreaks is a significant problem, especially with multidrug-resistant (MDR) TB strains [3,4].

Genotyping and molecular studies play an important role in investigating laboratory cross-contamination [5]. Cross-contamination is suspected when a TB strain matches the genotype of another isolate processed during the same period in a laboratory institute, without epidemiological links [6]. When undetected, cross-contamination results in unnecessary expense arising from contact investigation and drug treatments with serious side effects [7].

Whole-genome sequencing (WGS) shows promise in the quality control of *Mycobacterium tuberculosis* (MTB) laboratory cultures because it is faster and more accurate than other common methods [8] and can be applied to the identification of highly homologous strains, as well as to gene mutations for drug resistance [9,10,11].

We describe the clinical and laboratory characteristics of false positive versus true positive MTB cultures based on WGS, showing that genomics can be used in clinical practice and quality control.

## 2. Results

### 2.1. Whole-Genome Sequencing Data

Clinical information about the six patients is presented in Table 1. According to WGS analysis, five of the seven isolates were identical, while the remaining two isolates, both of which were from the sixth patient, differed from the others by 548 single-nucleotide polymorphisms (SNPs) (Figure 1). No SNPs were observed among isolates from patients No. 1 to No. 5 and between those from patient Nos. 6-1 and 6-2. In the phylogenetic tree, identical and different strains were distributed separately. Combining these results with clinical information, we found that the five identical strains resulted from laboratory cross-contamination, and the other two strains were true pre-extensive drug-resistant (pre-XDR) TB without epidemiological links (Figure 1 and Figure 2). Gene mutation results for the cross-contaminated samples showed resistance to isoniazid, rifampicin, and quinolone, but not to ethambutol and streptomycin, which differed from previous phenotypic drug susceptibility test (pDST) results on resistance to ethambutol and streptomycin. However, one of the sixth patient’s results coincided with their previous pDST results (Table 1).

### 2.2. Correlation with Epidemiological and Clinical Information

A detailed investigation showed that the first four patients had visited or had been admitted to the H hospital due to respiratory symptoms during the same period. Sputum acid-fast bacilli (AFB) culture and pDST from the H hospital patients were performed at the commercial lab company (SG) and supranational laboratory, respectively. The fifth patient was confirmed by the S hospital as pre-XDR-TB based on culture and pDST performed by the same laboratories. The sixth patient was confirmed to have pre-XDR-TB with the same pDST at the public health center, and the AFB culture was confirmed as positive at our hospital.

Most radiology findings showed minimal inflammatory lesions with or without old TB lesions (Table 2). All patients were receiving first-line TB drugs before the pDST results were obtained. The first patient died on 2 June. When the other five patients were transferred to our hospital, bronchoscopy washings were performed. We started pre-XDR medications (bedaquiline, clofazimine, cycloserine, prothionamide, and linezolid) for the fourth and sixth patients (linezolid, kanamycin, cycloserine, clofazimine, and levofloxacin). We prescribed no TB medication for the other three patients until we received the final AFB culture results. After receiving the WGS results, we stopped the pre-XDR medication for the fourth patient. Later, the results of the repetitive sputum AFB culture and bronchoscopic washing AFB culture performed in our hospital were also negative for the second, third, fourth, and fifth patients. We changed the medications to linezolid, cycloserine, clofazimine, protionamide, bedaquiline, and para-aminosalicylic acid for the sixth patient when the WGS patterns indicated that the patient truly had pre-XDR-TB.

## 3. Discussion

WGS aided in ruling out a pre-XDR-TB outbreak, revealing it to be a case of laboratory cross-contamination. By combining WGS results and clinical findings, we could conclude that the strains isolated from the second to fifth patients were contaminated with the strain infecting the first patient, and the sixth patient was another real TB patient infected with different strains. The usual mechanisms of cross-contamination with MTB include technician error, reagent contamination, and equipment failure [7,12]. We initially suspected contamination during specimen delivery; however, we eventually confirmed cross-contamination, possibly occurring during the sputum specimen process, in a large reference laboratory holding strains from many hospitals.

Mycobacterial interspersed repetitive unit–variable number tandem repeat (VNTR) genotyping has been the most common method for detecting cross-contamination [4,8,13]. However, because it only uses part of the genetic information [10], WGS might be a more informative method for identifying cross-contamination. WGS analysis indicated that the isolates of the sixth patient differed from those of the other patients with cross-contaminated strains. WGS corrected the erroneous pDST results of all patients.

In addition to using epidemiological and clinical data, the TB laboratory process should be monitored to reduce cross-contamination, which could result in unnecessary expenditure, especially in MDR-TB cases [14].

Our study has several strengths compared to a previous study [6]. First, to discriminate cross-contamination from an outbreak, we not only performed confirmatory laboratory tests but also considered the results of the repetitive sputum study, bronchoscopy, symptoms of TB, and imaging findings. Second, our sample included not only drug-susceptible or simple MDR but also pre-XDR-TB cases, which could be critical to the patients and community if the contamination or outbreak is not rapidly discriminated. Third, we performed genetic mutation tests to assess drug resistance, which enabled rapid decision making regarding the choice of medication in drug-resistant cases, as well as SNP analysis by WGS.

There were limitations in tracing all possible cross-contaminated isolates because the use of data without patient information was not possible. The treatment of a true positive pre-XDR patient, the sixth patient, was delayed by the process of ruling out cross-contamination.

## 4. Materials and Methods

### 4.1. Correlation with Epidemiological and Clinical Information

For clinical information, we collected information regarding age, sex, underlying disease, symptoms, past TB history, chest CT findings, bronchoscopic findings (if performed), sputum and/or bronchoscopic washing, AFB stain, TB culture and pDST results, and the treatment regimen of TB from medical records.

Initially, four patients were transferred from the H hospital. Suspecting a TB outbreak with pre-XDR strains based on their medical histories, including the same pDST results, we requested the H hospital to check whether more pre-XDR-TB patients were confirmed during the same period. Subsequently, a fifth patient was transferred to our hospital from the S hospital, which is located in the same village, after confirming the presence of pre-XDR-TB with the same pDST results during the same period. One patient from the H hospital died of severe pre-XDR-TB. Hence, we considered the possibility of a serious TB outbreak in the H hospital or local community. AFB sputum specimens from these five patients had been delivered to an aforementioned SG for culture, and the cultured specimens were delivered to a supranational lab for pDST. The pDST results from all patients showed resistance to isoniazid, ethambutol, rifampicin, pyrazinamide, quinolone, and streptomycin. Subsequently, a sixth patient was transferred to our hospital and confirmed to have MDR-TB by a rapid pDST. We suspected that this patient might have been infected by index TB cases as well in the same village. We performed WGS for all six patients to investigate recent transmissions.

### 4.2. Whole-Genome Sequencing

Strain harvest and DNA extraction were performed as previously described [11]. The library was prepared using the Ion Xpress™ Plus Library Kit for the AB Library Builder™ System, according to the manufacturer’s instructions. The barcoded libraries were pooled for the following goal: a sequence coverage of approximately 100×; the template was prepared and loaded onto the Ion Chip 530 in the Ion Chef™ machine. WGS was performed using Ion GeneStudio S5 (Thermo Fisher Scientific, Madison, WI, USA). The FASTQ files of the sequences were introduced and analyzed using Bionumerics software v7.6 (Applied Maths, Kortrijk, Belgium). The program mapped the sequences against the reference strain H37Rv (GenBank accession no. NC_000962.3) and obtained different SNPs among the genomes analyzed. The SNP analysis allowed the construction of the dendrogram using the complete linkage method [15]. For rapid antibiotic resistance prediction, the Mykrobe Predictor TB (v0.1.3.) software was employed. We ran the program offline after installing it locally [16]. The study was approved by each institutional review board (2015AS0056, 2017GR0301, 2019-KNTA-IRB-02).

## 5. Conclusions

WGS, along with clinical information, may be useful to differentiate laboratory cross-contamination from TB outbreaks, thereby avoiding the unnecessary treatment of false positive patients and the delayed treatment of true positive TB patients by using reliable genotyping for drug resistance. Our study shows that identical isolate pairs must not be confirmed as a case of recent transmission without supporting epidemiological and clinical information.

## Figures and Tables

**Figure 1 antibiotics-10-00297-f001:**
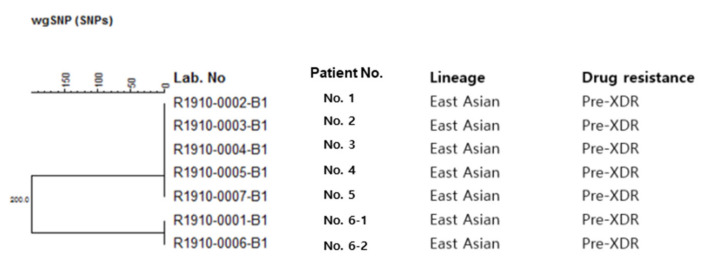
Phylogenetic tree of the patients.

**Figure 2 antibiotics-10-00297-f002:**
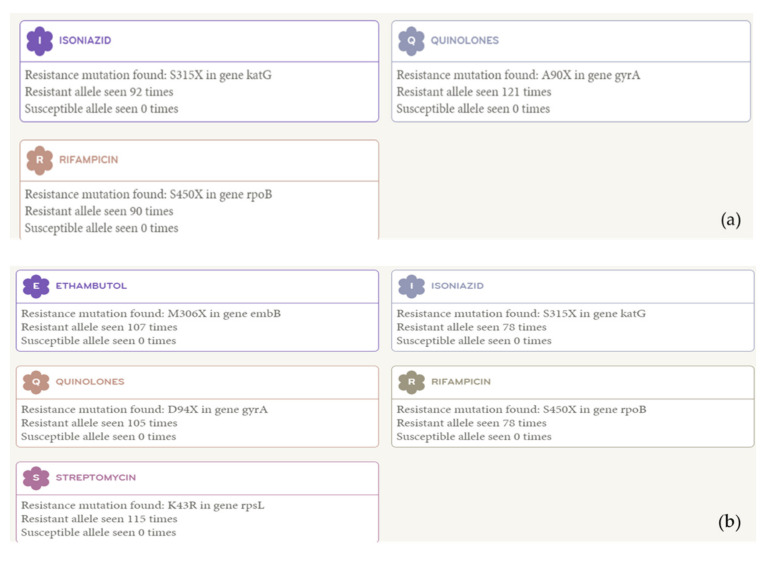
The gene mutation results of strains isolated from the patients. (**a**) The gene mutation results of the first, second, third, fourth, and fifth patients, (**b**) The gene mutation results of the sixth patient.

**Table 1 antibiotics-10-00297-t001:** True and false positive results of the AFB smear, AFB culture, and drug resistance reports by pDST and WGS.

Case No.	Smear Date	Smear Results *	Culture Date	Culture Results	Concordance of Strain	H	RFP	SM	E	Km	Cm	PTH	CS	PAS	Ofx	Mfx	Amk	Lfx	Rib	Z	LNZ
No. 1	21 May	−	21 May	1+	Concordant 1	R	R	R	R	S	S	S	S	S	R	R	S	R	R	R	S
	23 May	4+	23 May	4+																	
	24 May	4+	24 May	1+																	
	25 May	4+	25 May	1+																	
	26 May	4+	26 May	1+																	
	1st June	2+	1 June	1+																	
No. 1 WGS						R	R	S	S	S	S	S	S	S	R	R	S	R	R	R	S
No. 2	24 May	−	25 May	1+		R	R	R	R	S	S	S	S	S	R	R	S	R	R	R	S
No. 2. WGS						R	R	S	S	S	S	S	S	S	R	R	S	R	R	R	S
No. 3	27 May	−	27 May	2+		R	R	R	R	S	S	S	S	S	R	R	S	R	R	R	S
No. 3 WGS						R	R	S	S	S	S	S	S	S	R	R	S	R	R	R	S
No. 4	24 May	−	24 May	1+		R	R	R	R	S	S	S	S	S	R	R	S	R	R	R	S
No. 4 WGS						R	R	S	S	S	S	S	S	S	R	R	S	R	R	R	S
No. 5	27 May	−	27 May	2+		R	R	R	R	S	S	S	S	S	R	R	S	R	R	R	S
No. 5 WGS						R	R	S	S	S	S	S	S	S	R	R	S	R	R	R	S
No. 6-1 (sputum)	2 June	+	26 June	1+	Concordant 2	R	R	R	R	S	S	S	S	S	S	S	S	S	R	R	S
No. 6-1 WGS						R	R	R	R	S	S	S	S	S	R	R	S	R	R	R	S
No. 6-2 (bfs)	4 July	2+	4 July	1+		R	R	R	R	S	S	S	S	S	R	R	S	R	R	R	S
No. 6-2 WGS						R	R	R	R	S	S	S	S	S	R	R	S	R	R	R	S

* The results of AFB smears were graded according to the American Thoracic Society/Center for Disease Control and Prevention (ATS/CDC) as follows: −, no bacilli in 300 fields; +1, 1–9 bacilli in 100 fields; +2, 1–9 bacilli in 10 fields; +3, 1–9 bacilli in one field; and +4, >9 bacilli in one field. Smear and culture results of the first to fifth patients are all sputum results. AFB, Acid-Fast Bacilli; pDST, phenotypic drug susceptibility testing; WGS, whole-genome sequencing; bfs, bronchofiberscopy; H, isoniazid; RFP, rifampin; SM, streptomycin; E, ethambutol; Km, kanamycin; Cm, capreomycin; PTH, prothionamide; CS, cycloserine; PAS, para-aminosalicylic acid; Ofx, ofloxacin; Mfx, moxifloxacin; Amk, amikacin; Lfx, levofloxacin; Rib, rifabutin; Z, pyrazinamide; LNZ, linezolid; S, susceptible; R, resistant.

**Table 2 antibiotics-10-00297-t002:** Clinical information of six patients with positive *Mycobacterium tuberculosis* culture as analyzed at a private laboratory company.

Case No.	Age/Sex	Underlying Disease	Symptoms	Past TB History	Chest CT Findings	Bronchoscopy	TB Treatment		Clinical Diagnosis	Laboratory Investigation
							Previous Hospital	After Referral		
**1**	58/Male	Chronic alcoholics	Fever, anorexia	−	Multiple cavities and consolidation in both lungs	Not done	Yes	Expired	Active TB	True strain
**2**	69/Male	DM, HTN, Cerebral infarction	Cough, sputum	+	Multiple centrilobular GGOs and nodules in RML/RLL	No endobronchial lesion	Yes	No	CAP	Contamination
**3**	73/Female	HTN, Spinal stenosis	Cough, sputum	−	Diffuse ill-defined centrilobular nodules and GGO in both lungs	Anthracofibrosis	Yes	No	Bronchitis	Contamination
**4**	71/Male	HTN, Asthma, Unstable angina, CKD, Panic disorder	General weakness	+	Multiple calcified granulomas, irregular pleural thickening, and bronchiectasis	Anthracofibrosis	Yes	Yes	Bronchitis	Contamination
**5**	32/Female	none	Cough, sputum	+	Consolidation, calcified granulomas, fibrosis, and pleural thickening in both lungs	Anthracotic pigmentation	Yes	No	CAP	Contamination
**6**	55/Female	HTN	Cough, sputum	−	Multiple ill-defined centrilobular nodules and patchy consolidations in LUL	Endobronchial TB (left main)	Yes	Yes	Active TB	True strain

DM, diabetes mellitus; HTN, hypertension; CKD, chronic kidney disease; TB, tuberculosis; CT, computed tomography; GGO, ground-glass opacity; LUL, left upper lobe; RML, right middle lobe; RLL, right lower lobe; CAP, community-acquired pneumonia.

## Data Availability

All datasets were provided by requests to the author for correspondence.

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
