# Peer review of "Cross-Contamination versus Outbreak: Pre-XDR Mycobacterial Strains Confirmed by Whole-Genome Sequencing"

_antibiotics, 2021, doi:10.3390/antibiotics10030297_

Round 1
Reviewer 1 Report
Manuscript: Cross-contamination versus outbreak: Pre-XDR mycobacterial strains confirmed by whole-genome sequencing, presents short communication about false-positive versus true-positive MTB isolates obtained by WGS. Overall, the study is not very insightful as most of the presented results are expected based on what is already known regarding the problem of common laboratory cross-contamination (also discussed and presented in literature cited by authors). I do not find novelty in described cases nor in the methods exploited.
I do not recommend manuscript for publication mainly due to lack of novelty.
Author Response
We appreciate your review and comments.

Reviewer 2 Report
The authors reported false-positive cases of TB which might have occurred due to laboratory contamination. Based on WGS technology, they decipher the Mtb strains and concluded that careless handling and processing of the samples might have wrongly reported and an outbreak of TB which is alarming.
My concerns are as follows:
- Table 2 does not include the AFB smear and AFB culture of each patient. Did the authors perform a similar test to claim contamination?
- Can the authors have access to the old samples in the previous hospital and validate the WGS?
- if the authors recommend WGS as a method to diagnose, I wonder how cost-effective would it be for poor patients' families?
Author Response
<Response to reviewer 2>
The authors reported false-positive cases of TB which might have occurred due to laboratory contamination. Based on WGS technology, they decipher the Mtb strains and concluded that careless handling and processing of the samples might have wrongly reported and an outbreak of TB which is alarming.
My concerns are as follows:
Q1> Table 2 does not include the AFB smear and AFB culture of each patient. Did the authors perform a similar test to claim contamination?
A1> We included the AFB smear and AFB culture results in Table 1. The results included in Table 1 are only the positive results. We repeatedly performed AFB smears, culture, and bronchoscopic washing for patients No. 2 to 5 after referral to our hospital, and the results were all negative.
We added this in line L99-101
Later, the results of repetitive sputum AFB culture and bronchoscopic washing AFB culture performed in our hospital were also negative for 2nd, 3rd, 4th and 5th patients.
We changed the title of table 1 to make it clear,
Results of the AFB smear, AFB culture, and drug resistance reports by pDST and WGS
->True and false Positive results of the AFB smear, AFB culture, and drug resistance reports by pDST and WGS
Q2> Can the authors access the old samples in the previous hospital and validate the WGS?
A2> We have access to both samples from SG (H hospital referred the AFB culture to the SG company) and supranational laboratory (H hospital referred DST to a supranational laboratory).
We described this in line 84-86”
“Sputum acid-fast bacilli (AFB) culture and pDST from the H hospital patients were performed at the commercial lab company (SG) and supranational laboratory, respectively.”
Q3> if the authors recommend WGS as a method to diagnose, I wonder how cost-effective would it be for poor patients' families?
A2> Thank you for your insightful thoughts and comments. We agree that this is an important point, and we need to investigate the cost-effectiveness of using WGS as a diagnostic method.
A previous study reported that the price for each patient was $63 (ref: He, G et al. Prediction of treatment outcomes for multidrug-resistant tuberculosis by whole-genome sequencing. International Journal of Infectious Diseases, 2020, 96: 68-72.) Moreover, another study commented that the investment in routine DST for all TB patients using WGS may not seem cost-effective in the short term, but studies on the impact and cost effectiveness of routine WGS in a high-burden setting are needed to determine the feasibility of WGS. WGS analysis yields susceptibility results for both first- and second-line drugs; these data could also be an invaluable resource for understanding the epidemiology of TB. (ref: Iketleg, T et al. Mycobacterium tuberculosis next-generation whole-genome sequencing: opportunities and challenges. Tuberculosis Research and Treatment, 2018, 2018.)

Reviewer 3 Report
The paper describes whole genome sequencing for 7 strains of Mycobacterium tuberculosis isolated from 6 patients. The authors used the whole genome sequences to examine whether an outbreak of tuberculosis occurred. It was not the outbreak but false-positive due to cross-contamination. The authors concluded that whole genome sequencing is useful to differentiate cross-contamination from true-positive in clinical tuberculosis infections.
This study provides readers with useful information. The manuscript is well written, but the results may be hard to interpret stated as my ‘Major comment’.
Major comment
(1) What does ‘cross-contamination’ mean in this study? Is the following interpretation correct?
-The strains isolated from 1st to 5th patients had not infected to each patient but were contamination of the strain infected in 6th patient.
If so, I cannot understand why the resistance for SM and E are different between pDST and WGS. At my first reading, I thought that genomic DNA samples had been contaminated (genomic DNA samples of 1st to 5th patients were replaced with that of an isolate from the other patient). What do the authors think the possibility of cross-contamination (replacement) of DNA samples. Please explain the authors’ interpretation not to make readers to misunderstand as I did.
(2) L110-111. Why can the authors conclude that results by WGS are correct and those by pDST are in error?
(3) Materials and Methods. Please include all the method to obtain data in Figures 1 and 2 and Table 1.
(4) In this study, 2nd to 5th patients were not infected by M. tuberculosis and
strains isolated from them were derived from to cross-contamination. If the 1st to 5th patients had been actually infected due to an outbreak, what would the results be? In general, is an outbreak caused by a single strain, isn’t it? Is
so, the same strain could be isolated from 1st to 5th patients. Hence, no SNP
would be observed like Fig 1. Does this mean that WGS is not useful to clarify
if it is cross-contamination or an actual outbreak? Please tell me what results
will be in the following cases: (i) there are neither cross-contamination nor
an outbreak. (I guess any M. tuberculosis strains will not be isolated from the
patients); (ii) there is not cross-contamination but an outbreak. (Will the
result be the same as Fig 1 without SNPs?); (iii) there is not an outbreak but
cross-contamination. (The present results); (iv) there are both of cross-contamination and an outbreak (Can the authors conclude that the both occurred?).
Minor comments
- L24-25. It is unclear what groups C and A are.
- LL53-54. The maximum number of SNPs between any pair of isolates was 0 among the concordant specimens. -> No SNPs was observed among isolates from patient No. 1 to No. 5 and between those from patient No. 6-1 and No. 6-2, respectively.
- ‘, which were’ -> ‘on’
- Legend of Fig 1. results of the -> results of strains isolated from the
- Table 1. It is unclear what ‘AFB’, ‘Smear time’, smear results such as -, 4+ and 2+, and culture results such as 1+,2+ and 3+ mean. The data of ‘Culture time’ is not time but date. It is unclear which R means, rifampin or resistant. Do not the authors need to show sputum or bfs for Cases No. 1 to 5?
- prohibited?
- DST -> pDST
Author Response
<Response to reviewer 3>
The paper describes whole genome sequencing for 7 strains of Mycobacterium tuberculosis isolated from 6 patients. The authors used the whole genome sequences to examine whether an outbreak of tuberculosis occurred. It was not the outbreak but false-positive due to cross-contamination. The authors concluded that whole genome sequencing is useful to differentiate cross-contamination from true-positive in clinical tuberculosis infections.This study provides readers with useful information. The manuscript is well written, but the results may be hard to interpret stated as my ‘Major comment’.
Major comment
Q1> What does ‘cross-contamination’ mean in this study? Is the following interpretation correct? The strains isolated from 1st to 5th patients had not infected to each patient but were contamination of the strain infected in 6th patient. If so, I cannot understand why the resistance for SM and E are different between pDST and WGS. At my first reading, I thought that genomic DNA samples had been contaminated (genomic DNA samples of 1st to 5th patients were replaced with that of an isolate from the other patient). What do the authors think the possibility of cross-contamination (replacement) of DNA samples. Please explain the authors’ interpretation not to make readers to misunderstand as I did.
A1> We appreciate your comment. The strains isolated from the 2nd to 5th patients were infected with the strain infecting the 1st patient, and the 6th patient was another real TB patient with different strains.
We added the following sentence to clarify the meaning of our results in L112-115.
“By combining WGS results and clinical findings, we could conclude that the strains isolated from the second to fifth patients were contaminated with the strain infecting the first patient, and the sixth patient was another real TB patient infected with different strains.”
In addition, the result of the DST by WGS was susceptible, while that of DST by pDST in SM, E showed different results in the 1st patient. This finding implies that we need to find further genetic mutations relevant to the resistance to secondary medication.
Q2> L110-111. Why can the authors conclude that results by WGS are correct and those by pDST are in error?
A2> This is a good point. We appreciate your comment. As mentioned above, there is a chance that we may not find additional resistance-related genes using WGS. Thus, we need to further investigate other genetic mutations causing resistance to secondary medication. Also, there is a possibility that the pDST results for the secondary medication are inaccurate. Thus, the accuracy of pDST should be improved.
Q3> Materials and Methods. Please include all the method to obtain data in Figures 1 and 2 and Table 1.
A3> We added the details of all methods in the Methods section.
L134-137 we added,
For clinical information, we collected information regarding the age, sex, underlying disease, symptoms, past TB history, chest CT findings, bronchoscopic findings (if performed), sputum and/or bronchoscopic washing, AFB stain, TB culture, and pDST results, and the treatment regimen of TB from medical records.
L154-166, we added and changed the sentences.
WGS was performed using Ion GeneStudio with an ION530tm chip (Thermo Fisher Scientific, Madison, WI, USA), and the average sequencing depth was approximately 100-fold.
->The library was prepared using the Ion Xpress™ Plus Library Kit for AB Library Builder™ System according to the manufacturer’s instructions. The barcoded libraries were pooled for the following goal: a sequence coverage of approximately 100×; the template was prepared and loaded onto the Ion Chip 530 in the Ion Chef™ machine. WGS was performed using the Ion GeneStudio S5 (Thermo Fisher Scientific, Madison, WI, USA). The fastq files of the sequences were introduced and analyzed using Bionumerics software v7.6 (Applied Maths, Kortrijk, Belgium). The program mapped the sequences against the reference strain H37Rv (GenBank accession no. NC_000962.3) and obtained different SNPs among the genomes analyzed. The SNP analysis allowed the construction of the dendrogram using the complete linkage method [15]. For rapid antibiotic resistance prediction, the Mykrobe Predictor TB (v0.1.3.) software was employed. We ran the program offline after installing it locally [16].
Q4> In this study, 2nd to 5th patients were not infected by M. tuberculosis and strains isolated from them were derived from to cross-contamination. If the 1st to 5th patients had been actually infected due to an outbreak, what would the results be? In general, is an outbreak caused by a single strain, isn’t it? Is so, the same strain could be isolated from 1st to 5th patients. Hence, no SNP
would be observed like Fig 1. Does this mean that WGS is not useful to clarify
if it is cross-contamination or an actual outbreak?
A4> By using WGS, we can quickly determine whether the strain is identical. Thus, by using WGS, we can save time and be prepared for contact investigation when an outbreak is suspected. WGS also provides information about the resistance pattern of the medication, and the community can be prepared for further steps. However, as you mentioned, we cannot discriminate between true infection and contamination using WGS results alone. As we have described in the manuscript, contamination should be judged by additional clinical information such as repetitive sputum study results, bronchoscopy, symptoms, and imaging findings.
Q5>Please tell me what results will be in the following cases:
A5>
(i) there are neither cross-contamination nor an outbreak. (I guess any M. tuberculosis strains will not be isolated from the patients);
Yes, this is correct. M. tuberculosis strains will not be isolated from the patients.
(ii) there is not cross-contamination but an outbreak. (Will the result be the same as Fig 1 without SNPs?);
Yes, the results of the WGS would be identical, and the clinical symptoms and images of all patients should be compatible with pulmonary TB.
(iii) there is not an outbreak but cross-contamination. (The present results);
The results of the WGS would be identical, but the clinical symptoms and images of patients would not be compatible with TB, and the repetitive microbiological results would be negative.
(iv) there are both of cross-contamination and an outbreak (Can the authors conclude that the both occurred?).
In this case, it would be interpreted as an outbreak, because patients might show clinical findings compatible with TB, and the same strain might be reported using WGS. However, if there is a possibility of contamination above the range of the outbreak, we should look at the clinical evidence for real TB using clinical findings.
Minor comments
Q1> L24-25. It is unclear what groups C and A are.
A1> For clarity, we changed the sentences as below
“Further, the results of WGS gene-mutation analysis for ethambutol, streptomycin, and fluoroquinolone resistance in all six patients were not concordant with the pDST results”
Q2> LL53-54. The maximum number of SNPs between any pair of isolates was 0 among the concordant specimens. -> No SNPs was observed among isolates from patient No. 1 to No. 5 and between those from patient No. 6-1 and No. 6-2, respectively.
A2> We have changed this sentence per your recommendation in L55-56
Q3> ‘, which were’ -> ‘on’
A3> We changed the text per your recommendation in L63.
Q4> Legend of Fig 1. results of the -> results of strains isolated from the
A4> Per your recommendation, we changed the legend of Fig. 2.
Q5> Table 1. It is unclear what ‘AFB’, ‘Smear time’, smear results such as -, 4+ and 2+, and culture results such as 1+,2+ and 3+ mean.
A5>We added sentences below the table 1.
*The results of AFB smears were graded according to the American Thoracic Society/Center for Disease Control and Prevention (ATS/CDC) as follows: -, no bacilli in 300 fields; +1, 1–9 bacilli in 100 fields; +2, 1–9 bacilli in 10 fields; +3, 1–9 bacilli in one field; and +4, >9 bacilli in 1 field.
Q6>The data of ‘Culture time’ is not time but date.
A6> Per your recommendation, we changed the word “time” to “date” in Table 1.
Q7> It is unclear which R means, rifampin or resistant.
A7> We changed in the abbreviation of rifampin to ‘RFP’ to address this issue.
Q8>Do not the authors need to show sputum or bfs for Cases No. 1 to 5
A8> For patients No. 1 to 5, we did not perform BFS before referral to our hospital. Patients No. 2–5 were all examined by BFS after referral to our hospital; the results for these patients were negative.
Therefore, we have inserted the following footnote prevent misunderstanding.
*The smear and culture results of the 1st to 5th patients are all sputum results.
The results included in Table 1 are only the positive results. We repeatedly performed AFB smears, culture, and bronchoscopic washing for patients No. 2 to 5 after referral to our hospital, and the results were all negative.
We added this in line L99-101
Later, the results of repetitive sputum AFB culture and bronchoscopic washing AFB culture performed in our hospital were also negative for 2nd, 3rd, 4th and 5th patients.
We changed the title of table 1 to make it clear,
Results of the AFB smear, AFB culture, and drug resistance reports by pDST and WGS
->True and false positive results of the AFB smear, AFB culture, and drug resistance reports by pDST and WGS
Q9> prohibited?
A9> We changed the word prohibited to ‘was not possible’ in L129.
Q10> DST -> pDST
A10> We changed the text per your recommendation in L150.
We further edited some minor grammar errors and marked the changes in red.

Round 2
Reviewer 1 Report
I am disappointed that Authors did not even make an effort to discuss about my main critical comment. I still think that the manuscript lack novelty. The only answer I got is: “Thank you for your critical comments…” and I see no explanation that could support their idea of publication-not in the text nor in the Author’s reply. Therefore I do not feel that Authors have strong confidence of their idea of research and publication. I still do not recommend manuscript for publication due to lack of novelty.
Author Response
We apologize that we did not provide any explanation on your comment.
As you mentioned, our study is not the first to confirm cross-contamination by WGS. However, compared to a previous study (Wu, J; Yang, C; Lu, L; Dai, W. Detection of tuberculosis laboratory cross-contamination using whole-genome sequencing. Tuberculosis (Edinb) 2019, 115, 121–125.), we believe that our work is novel in three ways.
First, to discriminate cross-contamination from an outbreak, we not only performed confirmatory laboratory tests but also considered symptoms compatible with TB, chest CT findings, and repetitive sputum study including bronchoscopic washing. These have not been performed in previous studies on cross-contamination. It is possible to rapidly determine identical strains by WGS. Thus, we can save time by performing WGS and prepare for contact investigation when an outbreak is suspected. However, we could not discriminate between true infection and contamination based on results of WGS alone. Contamination should be confirmed based on additional information such as results of the repetitive sputum study, bronchoscopy, symptoms, and imaging findings. By combining the results of WGS and clinical findings, we concluded that the strains isolated from the second to fifth patients were contaminated with the strain infecting the first patient, and the sixth patient was another real TB patient infected with different strain.
Second, our sample included not only drug-susceptible or simple MDR but also pre-XDR TB cases, which could be critical to the patients and community if the contamination or outbreak is not rapidly discriminated. We conducted this study not just as a laboratory investigation. We suspected when several pre-XDR cases were sequentially referred to our hospital and we synthetically analyzed the clinical findings and WGS data for clinical decision. Thus, we learned that suspicion is important, and the combination of WGS and clinical findings could prevent unnecessary aggressive treatment and isolation of patients.
Third, we performed genetic mutation tests and SNP analysis. WGS also provides information regarding the drug resistance pattern, which can be helpful to the community to plan further steps, especially in drug-resistant cases. Genetic mutation tests of resistant genes can aid in rapid decision-making (2 weeks) regarding the choice of medication in real cases. Currently, patients have to wait for more than 2 months for results of pDST. However, in our study, the result of the pDST by WGS was susceptible, while that of DST by pDST in SM and E showed different results in the 1st patient. There is a possibility that additional resistance-related genes may not be identified by WGS. Therefore, other genetic mutations that cause resistance to secondary medications must be investigated. In addition, there is a possibility of inaccurate results of pDST for secondary medications. In the sixth patient, the resistance pattern assessed by pDST was different in the strains obtained from sputum and bronchoscopic washing (Mfx S in sputum and R in bronchoscopic washing). However, gene mutation patterns analyzed by WGS were identical in both sputum and bronchosopic washing in the same patient (Mfx R in sputum and bronchoscopic washing). This indicates that the accuracy of conventional pDST requires improvement.
We have emphasized the strength and novelty of our study in the Discussion section (Lines 127-133) as follows:
Our study has several strengths compared to a previous study [6]. First, to discriminate cross-contamination from an outbreak, we not only performed confirmatory laboratory tests but also considered the results of the repetitive sputum study, bronchoscopy, symptoms of TB, and imaging findings. Second, our sample included not only drug-susceptible or simple MDR but also pre-XDR TB cases, which could be critical to the patients and community if the contamination or outbreak is not rapidly discriminated. Third, we performed genetic mutation tests to assess drug resistance, which enabled rapid decision-making regarding the choice of medication in drug-resistant cases, as well as SNP analysis by WGS.
We appreciate your comments and thank you for providing us with the opportunity to refine our manuscript.

Reviewer 2 Report
The authors addressed the questions and made corrections wherever necessary. They clarified the AFB staining and corrected the table.
They also agreed about the limitation of the study with the cost of the WGS if they propose to use it as a diagnostic.
Reviewer 3 Report
The authors adequately responded and revised the manuscript.